# Phospholipid Acyltransferases: Characterization and Involvement of the Enzymes in Metabolic and Cancer Diseases

**DOI:** 10.3390/cancers16112115

**Published:** 2024-05-31

**Authors:** Jan Korbecki, Mateusz Bosiacki, Maciej Pilarczyk, Magdalena Gąssowska-Dobrowolska, Paweł Jarmużek, Izabela Szućko-Kociuba, Justyna Kulik-Sajewicz, Dariusz Chlubek, Irena Baranowska-Bosiacka

**Affiliations:** 1Department of Anatomy and Histology, Collegium Medicum, University of Zielona Góra, Zyty 28, 65-046 Zielona Góra, Poland; jan.korbecki@onet.eu; 2Department of Biochemistry and Medical Chemistry, Pomeranian Medical University in Szczecin, Powstańców Wlkp. 72, 70-111 Szczecin, Poland; mateusz.bosiacki@pum.edu.pl (M.B.); dchlubek@pum.edu.pl (D.C.); 3Department of Nervous System Diseases, Neurosurgery Center University Hospital in Zielona Góra, Collegium Medicum, University of Zielona Gora, 65-417 Zielona Góra, Poland; m.pilarczyk@inm.uz.zgora.pl (M.P.); p.jarmuzek@cm.uz.zgora.pl (P.J.); 4Department of Cellular Signalling, Mossakowski Medical Research Institute, Polish Academy of Sciences, Pawińskiego 5, 02-106 Warsaw, Poland; mgassowska@imdik.pan.pl; 5Institute of Biology, University of Szczecin, 13 Wąska, 71-415 Szczecin, Poland; izabela.szucko-kociuba@usz.edu.pl; 6Department of Conservative Dentistry and Endodontics, Pomeranian Medical University in Szczecin, Powstańców Wlkp. 72, 70-111 Szczecin, Poland; justyna.kulik.sajewicz@pum.edu.pl

**Keywords:** fatty acid, arachidonic acid, phospholipids, triacylglycerol, docosahexaenoic acid, 1-acylglycerol-3-phosphate acyltransferases, glycerol-3-phosphate acyltransferases, lysophospholipid acyltransferase

## Abstract

**Simple Summary:**

This review discusses the enzymatic processes governing the initial stages of the synthesis of glycerophospholipids (phosphatidylcholine, phosphatidylethanolamine, and phosphatidylserine) and triacylglycerol. The key enzymes analyzed include glycerol-3-phosphate acyltransferases (GPAT) and 1-acylglycerol-3-phosphate acyltransferases (AGPAT). Additionally, because most AGPATs have lysophospholipid acyltransferase (LPLAT) activity, enzymes involved in the Lands cycle with similar functions were also included. The review further explores the potential therapeutic implications of inhibiting these enzymes in the treatment of metabolic diseases and cancer. By elucidating the enzymatic pathways involved in lipid synthesis and their impact on various pathological conditions, the article contributes to the understanding of these processes and their potential as therapeutic targets.

**Abstract:**

This review delves into the enzymatic processes governing the initial stages of glycerophospholipid (phosphatidylcholine, phosphatidylethanolamine, and phosphatidylserine) and triacylglycerol synthesis. The key enzymes under scrutiny include GPAT and AGPAT. Additionally, as most AGPATs exhibit LPLAT activity, enzymes participating in the Lands cycle with similar functions are also covered. The review begins by discussing the properties of these enzymes, emphasizing their specificity in enzymatic reactions, notably the incorporation of polyunsaturated fatty acids (PUFAs) such as arachidonic acid and docosahexaenoic acid (DHA) into phospholipids. The paper sheds light on the intricate involvement of these enzymes in various diseases, including obesity, insulin resistance, and cancer. To underscore the relevance of these enzymes in cancer processes, a bioinformatics analysis was conducted. The expression levels of the described enzymes were correlated with the overall survival of patients across 33 different types of cancer using the GEPIA portal. This review further explores the potential therapeutic implications of inhibiting these enzymes in the treatment of metabolic diseases and cancer. By elucidating the intricate enzymatic pathways involved in lipid synthesis and their impact on various pathological conditions, this paper contributes to a comprehensive understanding of these processes and their potential as therapeutic targets.

## 1. Introduction

Glycerophospholipids and triacylglycerols (TAGs) undergo de novo synthesis from glycerol-3-phosphate, fatty acyl-CoA, and, in the case of phosphatidylethanolamine (PE) and phosphatidylcholine (PC), from cytidine diphosphate (CDP)-ethanolamine and CDP-choline, respectively [1]. The initial steps involve GPAT converting glycerol-3-phosphate and fatty acyl-CoA to lysophosphatidic acid (lysoPA) (also referred to as 1-acylglycerol-3-phosphate) (Figure 1) [2]. Subsequently, AGPAT catalyzes the formation of 1,2-diacylglycerol-3-phosphate (phosphatidic acid) [3]. This compound serves as a precursor for glycerophospholipids and TAG synthesis. The pathway then bifurcates into two routes: one leading to the synthesis of PC, PE, and phosphatidylserine (PS) via the Kennedy pathway and the other involving the production of phosphatidylinositol (PI), phosphatidylglycerol (PG), and cardiolipin (CL) after the conversion of phosphatidic acid to CDP-diacylglycerol (DAG) [4].

Glycerophospholipids can be synthesized through the Lands cycle, involving the de-esterification and re-esterification of phospholipids [5,6]. This cycle, facilitated by phospholipase A_2_ (PLA_2_) and LPLAT, introduces fatty acids into glycerophospholipids not incorporated by GPAT and AGPAT. The Lands cycle also plays a role in altering the lipid composition in response to environmental signals.

Glycerophospholipids, integral to cellular and intracellular membranes, serve a structural function. Changes in their composition can impact the properties of lipid rafts, influencing signal transduction from membrane receptors. Additionally, glycerophospholipids containing arachidonic acid participate in eicosanoid production, contributing to intercellular signaling.

TAG, composed of three fatty acids and glycerol, constitute the primary building blocks of lipid droplets, serving as the cell’s fat storage. These lipid droplets provide a hydrophobic environment, enabling the storage of poorly water-soluble substances, including lipophilic drugs [7].

The metabolism of glycerophospholipids and TAG is significant in diseases such as obesity and diabetes. Obesity, associated with excessive TAG accumulation in adipose tissue, affects a significant portion of the population in developed countries. Overweight is estimated to affect nearly two-thirds of the population in developed countries, for example, 64% in the United States [8]. Being overweight has a negative impact on health—it is strongly associated with insulin resistance [9,10,11] and connected with the predisposition toward certain types of cancer [12], including liver cancer [13], colorectal cancer [14], esophageal adenocarcinoma, endometrial cancer, gallbladder cancer, and renal cancer [15]. Obesity is also associated with poorer prognosis in cancer patients [16]. This demonstrates the important influence of metabolism on cancer processes.

This review aims to consolidate the existing knowledge on the connection between the initial steps of glycerophospholipid and TAG synthesis and cancer, considering the significant impact of metabolic disorders on cancer incidence and prognosis. Utilizing the Gene Expression Profiling Interactive Analysis (GEPIA) portal [17], a bioinformatic analysis was conducted on selected genes to assess their relevance to cancer processes [18].

To enhance the reader’s understanding, comprehensive information on the properties of the enzymes involved in glycerophospholipid and TAG synthesis has been compiled. Additionally, this review delves into the effects of these enzymes on non-cancerous diseases, including obesity and insulin resistance. This broader perspective sheds light on the intricate interplay between metabolic processes and health outcomes.

Furthermore, this review explores the potential for drug development targeting these enzymes. It provides insights into the current state of knowledge regarding the properties of enzyme inhibitors as potential therapeutic agents. The aim is to contribute to a better understanding of the molecular connections between metabolic pathways and cancer, offering insights into potential therapeutic avenues and emphasizing the need for continued research in this critical area.

## 2. Glycerol-3-Phosphate Acyltransferases

Upon the uptake or completion of fatty acid synthesis, these fatty acids are subsequently utilized for the production of complex lipids. The initial step in glycerophospholipid and TAG synthesis involves the generation of lysoPA (also referred to as 1-acylglycerol-3-phosphate) from glycerol-3-phosphate and fatty acyl-CoA. The enzymes responsible for catalyzing this reaction are GPAT [2]. In humans, four GPAT isoforms are discernible. Among these, glycerol-3-phosphate acyltransferase, mitochondrial (GPAM) (another name for GPAT1) [19], and GPAT2 [20] stand out as the key substances in glycerophospholipid synthesis. GPAM and GPAT2 are mitochondrial enzymes, so after lysoPA is biosynthesized, it is transported to the endoplasmic reticulum, where subsequent steps in the synthesis of glycerophospholipids and TAG take place. GPAM preferably utilizes saturated fatty acyl-CoAs such as palmitoyl-CoA C16:0 and lauroyl-CoA C12:0, as well as other fatty acyl-CoAs such as stearoyl-CoA C18:0, oleoyl-CoA C18:1n-9, and linoleoyl-CoA 18:2n-3 in smaller amounts [19,21]. This enzyme is significant in TAG synthesis [21,22]. GPAT2 does not show a preference for specific fatty acyl-CoAs [20].

The activity of GPAM can undergo regulation through changes in enzyme expression and phosphorylation events. This mechanism for increasing GPAM activity has been observed in mouse T-cell activation [23], in insulin action on rat adipocytes [24], and in studies of rat liver mitochondria [25].

Casein kinase 2 has been identified as a regulator of GPAM activity. Specifically, phosphorylation by casein kinase 2, observed at residues Ser^632^ and Ser^639^, has been shown to increase GPAM activity [24]. Additionally, another kinase known to increase GPAM activity through phosphorylation is protein kinase C (PKC)θ. This effect has been demonstrated in studies involving T lymphocytes [26]. In the context of adipocytes, insulin action has been linked to the phosphorylation of microsomal GPAT3 and GPAT4, leading to an increase in the activity of these enzymes. This highlights the intricate regulatory network influencing GPAM activity in various cellular processes [27].

Another two GPAT enzymes are GPAT3 (other names: AGPAT10, AGPAT9, and lysophosphatidic acid acyltransferase (LPAAT)-θ) [28] and GPAT4 [29], both of which are localized within the endoplasmic reticulum. Notably, it appears that GPAT3 may not play a significant role in glycerophospholipid production at the lysoPA synthesis stage but rather functions as an AGPAT [28]. However, another study showed that microsomal GPAT3 may be the major enzyme with GPAT activity in adipocytes [27].

The other names for GPAT4 are AGPAT6 and LPAAT-ζ [30]. GPAT4, besides AGPAT activity, also exhibits GPAT activity and is also known as GPAT4 [21]. GPAT4 is found in the endoplasmic reticulum and on lipid droplets [29] and prefers to utilize various fatty acyl-CoAs, with the lowest activity towards stearoyl-CoA C18:0 among the most commonly found acyl-CoAs in the cell [21]. While this enzyme is not significant in the synthesis of TAG and glycerophospholipids in adipose tissue and the liver, it is significant in the synthesis of PI and PC 34:1 [21,22]. In mammary epithelium, this enzyme is significant in the synthesis of TAG [31]. GPAT4 is present in lipid droplets and may be significant in TAG synthesis in these organelles [29].

GPAT plays a crucial role in TAG production, making it a key factor in the effects of high-fat diets [32]. Research on mice has demonstrated that GPAM, a specific isoform of GPAT, is implicated in insulin resistance in both muscle and the liver [33,34]. Moreover, GPAM is significant for TAG accumulation in the heart during high-fat diet conditions [35]. GPAT in mice on a high-fat diet has been shown to be responsible for increased DAG production in muscle. This compound is responsible for muscle insulin resistance [33]. GPAM may also be responsible for hepatic insulin resistance by participating in lipid production [34] and has been associated with liver cirrhosis [35]. GPAM is also a pro-cancer protein; in ovarian cancer, the expression of this protein is associated with a worse prognosis [36].

GPAT3 plays an important role in cancer, including colon cancer [37]. This enzyme causes the accumulation of lipid droplets in cancer cells. Thanks to this, such cancer cells will not activate the immune system after using standard anticancer drugs. Another isoform, GPAT4, has also been associated with insulin resistance. Studies involving *GPAT4* gene knockout mice fed a high-fat diet have revealed that GPAT4 is responsible for insulin resistance in the liver and muscle [38]. This emphasizes the critical role of GPAT enzymes in mediating the metabolic effects of high-fat diets and their impact on insulin sensitivity in various tissues.

## 3. Glycerol-3-Phosphate Acyltransferase Inhibitors

The first GPAT inhibitors, belonging to the o-(alkanesulfonamido)benzoic acid group, were developed in the early 2010s [39]. In vitro studies on isolated mitochondria demonstrated their inhibitory effects on mitochondrial GPATs, with half-maximal drug inhibitory concentrations (IC_50_) up to 17.4 μM [39]. Para-biphenyl analogs, particularly 4-([1,1′-biphenyl]-4-carbonyl)-2-(acetanesulfonamido)benzoic acid, have been identified as potent inhibitors within this compound class [40].

One of the well-known mitochondrial GPAT inhibitors is FSG67 (2-(nonylsulfonamido)benzoic acid) [41] (Figure 2). This compound shows an IC_50_ of 24.7 ± 2.1 μM on isolated mitochondrial GPATs [39]. On isolated mitochondria, FSG67 reduces the activity of GPAT with an IC_50_ of 30.2 μM and GPAT2 with an IC_50_ of 42.1 μM [41]. Research on diet-induced obese mice revealed that FSG67 in the dose of 5 mg/kg possesses anti-obesity properties [41]. FSG67 reduces food consumption, increases fat oxidation, and leads to decreased body fat. Prolonged exposure to FSG67 has been linked to increased insulin sensitivity, potentially attributed to its effects on muscle GPAT, known to contribute to insulin resistance in muscle [33,34].

FSG67’s impact on appetite is associated with its action on hypothalamic neurons, where it increases fatty acid oxidation, elevates ATP levels, reduces inflammatory responses, and consequently curtails appetite and food intake [42].

Beyond its role in treating obesity and related diabetes, FSG67 exhibits potential as an anticancer drug, particularly against acute myeloid leukemia (AML). AML is a type of leukemia that originates from hematopoietic stem cells [43]. AML cells, in comparison to normal hematopoietic stem/progenitor cells, exhibit increased GPAM expression and lysoPA production [44,45]. Inhibiting GPAM with FSG67 induces mitochondrial fission, decreases oxidative phosphorylation, and inhibits AML cell proliferation. Therefore, FSG67 has anti-leukemic properties against AML, which was confirmed by in vitro and in vivo studies on mice with inoculated AML [44]. Importantly, FSG67 demonstrates anti-leukemic properties against AML without affecting normal hematopoietic cells [44].

FSG67 also holds promise as a potential anticancer agent against other cancers. For instance, GPAM’s higher expression in ovarian cancer tumors is associated with a worse prognosis, making GPAM a plausible therapeutic target for ovarian cancer [46]. However, the suitability of GPAM as a therapeutic target varies among different cancer types; higher GPAM expression in breast cancer is associated with a better prognosis [46,47].

GPAM’s significance in immune system function indicates a potential side effect of using GPAT inhibitors. Studies on mice have revealed that GPAM in T lymphocytes is crucial for their function, particularly for their proliferation. The activation of T lymphocytes is followed by an increase in GPAM activity in these cells through phosphorylation of the enzyme [23]. At the same time, GPAM activation is reduced in old mice [23,26]. This is related to the decreased expression of acyl-CoA binding protein (ACBP) in T cells in older individuals [26], as well as mechanisms that inhibit activation by phosphorylation of this enzyme [26]. Reduced GPAM activity decreases Th1 cytokine production and increases Th2 cytokine production in T lymphocytes, which may be related to an increase in arachidonic acid incorporation into lipids and prostaglandin E_2_ (PGE_2_) production, as well as phospholipid production. All this suggests a role of GPAM in the regulation of immune responses [23,26,48].

## 4. 1-Acylglycerol-3-phosphate O-Acyltransferases and Lysophospholipids Acyltransferases

In the subsequent stage of glycerophospholipid synthesis, a second fatty acid is attached to lysoPA, resulting in the formation of phosphatidic acid (1,2-diacylglycerol-3-phosphate). The enzymes catalyzing this reaction are AGPATs [3]. In humans, there are eleven AGPAT enzymes: AGPAT1–11, mainly localized within the endoplasmic reticulum [3,49]. Notably, these enzymes exhibit pronounced substrate specificity for oleoyl-CoA C18:1n-9 [3,49,50,51]. As a result, glycerophospholipids in the sn-2 position predominantly contain monounsaturated fatty acids (MUFAs). However, not all AGPAT enzymes exhibit such substrate specificity. Lysocardiolipin acyltransferase (LCLAT)1 displays equal activity toward palmitoyl-CoA C16:0 and oleoyl-CoA C18:1n-9 [52].

### 4.1. AGPAT1

AGPAT1, also known as LPAAT-α and LPLAT1 [30], is situated in the endoplasmic reticulum [53]. Demonstrating AGPAT activity with significantly lower LPLAT activity [53], AGPAT1 facilitates the transfer of various fatty acids to lysoPAs, exhibiting heightened activity toward oleic acid C18:1n-9, linoleic acid C18:2n-6, and saturated fatty acids (SFA) with chain lengths of 14–16 carbons [53]. Vital for myoblast differentiation, AGPAT1’s role in cellular processes is underscored [54].

### 4.2. AGPAT2

Formerly known as LPAAT-β and LPLAT2 [30], AGPAT2 displays AGPAT activity with lower LPLAT activity [53]. AGPAT2 preferentially transfers oleic acid C18:1n-9 and linoleic acid C18:2n-6 to lysoPA, playing a pivotal role in adipogenesis [55]. Mutations in the AGPAT2 gene are linked to congenital generalized lipodystrophy type 1/Berardinelli-Seip congenital lipoatrophy type 1 (BSCL1), emphasizing its significance in lipid metabolism [55,56,57].

AGPAT2’s involvement in cancer is evident, with increased expression observed in ovarian cancer, correlating positively with cancer stage, grade, and mitotic index, indicating its potential prognostic value [58,59]. Additionally, AGPAT2 contributes to osteosarcoma tumorigenesis, and its expression is heightened under hypoxic conditions, promoting lipid droplet accumulation and chemoresistance [60,61]. AGPAT2 inhibitors developed in the early 2000s exhibited moderate antiproliferative properties in vitro on prostate and breast cancer cells, with in vivo studies demonstrating inhibitory effects on cancer tumor growth [62,63,64]. Despite these findings, further investigations into AGPAT2 inhibitors are warranted, with the potential for more promising inhibitors among the enzymes discussed in this section.

### 4.3. AGPAT3

AGPAT3, also identified as LPAAT3, LPAAT-γ, and LPLAT3 [30], sees upregulated expression driven by peroxisome proliferator-activated receptor (PPAR)δ and AMP-activated protein kinase (AMPK) [65]. Operating within the endoplasmic reticulum, AGPAT3 incorporates oleic acid C18:1n-9 and DHA into lysoPA, demonstrating LPLAT activity that specifically involves DHA in lysophospholipids [65,66]. AGPAT3 is essential for adipogenesis [67]. Reducing the expression inhibits the increase in fat tissue mass, which can be used in the treatment of obesity. AGPAT3 is also crucial for spermatogenesis [66] and influences myoblast differentiation, muscle response to exercise [65], and neuronal migration [68]. AGPAT3 has been associated with autism spectrum disorder [69]. Mutations in the AGPAT3 gene are linked to intellectual disability and retinitis pigmentosa syndrome (IDRP) [68].

AGPAT3’s dual role in cancer is evident, with increased expression noted in colorectal cancer tumors, while higher expression in gastric cancer correlates with a more favorable prognosis [70,71], suggesting potential anti-tumor properties.

### 4.4. AGPAT4

AGPAT4, recognized as LPAAT4, LPAAT-δ, and LPLAT4 [30], localizes in the endoplasmic reticulum and mitochondria [72,73]. Exhibiting AGPAT activity, AGPAT4 is responsible for incorporating PUFA, particularly DHA, into glycerophospholipids in the brain [72]. In colorectal cancer tumors, increased AGPAT4 expression leads to the release of lysoPA, activating macrophages and polarizing them into anti-tumor M1 macrophages, suggesting therapeutic potential in colorectal cancer [70].

### 4.5. AGPAT5

AGPAT5, also known as LPAAT-ε and LPLAT5 [30], resides not only in the endoplasmic reticulum but also in mitochondria, where it plays a role in attenuating fatty acid oxidation [49]. With a dual functionality, AGPAT5 exhibits LPLAT activity, contributing to the modification of fatty acid compositions within cell membrane phospholipids [49]. Notably, AGPAT5 displays a substrate preference, favoring the incorporation of oleic acid C18:1n-9 into lysoPA and DHA into lysophospholipids [49].

Contrary to some counterparts in the AGPAT family implicated in cancer, AGPAT5 diverges in its impact. Its expression diminishes in colorectal cancer, and higher levels are associated with a more favorable prognosis for patients with this cancer, suggesting an anticancer characteristic of AGPAT5 [74]. This unique profile positions AGPAT5 as a potential player in maintaining cellular homeostasis and underscores its relevance in the intricate landscape of lipid metabolism.

### 4.6. LPCAT1

Lysophosphatidylcholine acyltransferase 1 (LPCAT1) possesses AGPAT activity [50] and also exhibits activity towards lysophosphatidylcholine (lysoPC) and lysophosphatidylglycerol (lysoPG) [75]. Other names for this enzyme are AGPAT9 and LPLAT8 [19]. The highest expression of this enzyme is found in the lung and spleen [50]. In cells, LPCAT1 is localized in the endoplasmic reticulum. This enzyme has the highest substrate specificity for oleoyl-CoA [50]. For this reason, LPCAT1 protects against ferroptosis by reducing the amount of PUFA in the phospholipids of cell membranes [76]. This enzyme protects against the destructive effects of PUFA on the cell membrane and intracellular membranes. After synthesis or uptake by the cell, PUFA becomes part of the cell’s lipid membrane. Due to the spatial structure of lipids containing PUFA, structural changes occur, followed by damage to the cell’s intracellular membranes, particularly the endoplasmic reticulum, which activates the unfolded protein response (UPR) [77]. To counteract these effects, the cell produces dipalmitoylphosphatidylcholine, which mitigates the excessive impact of lipids containing PUFAs. The enzyme responsible for producing dipalmitoylphosphatidylcholine is LPCAT1 [77]. LPCAT1 is located in the nucleus [78] where it catalyzes histone H4 palmitoylation at Ser^47^ [79], which increases transcription. LPCAT1 and AGPAT11 are also present in lipid droplets [80], where they are responsible for the metabolism of lysoPC in the Lands cycle that involves the de-esterification and re-esterification of phospholipids [5,6].

LPCAT1 is involved in disease processes in psoriasis [81]. In psoriatic skin lesions, the expression of this enzyme is high. In keratinocytes, LPCAT1 increases nuclear factor-κB (NF-κB) activation and thereby glucose transporter 3 (GLUT3) expression. As a consequence, glycolysis, proliferation, and inflammatory reactions in the keratinocyte increase, which are an element of psoriasis. LPCAT1 has been implicated in various cancers, including hepatocellular carcinoma, breast cancer, head and neck squamous cell carcinoma, lung squamous cell carcinoma, lung adenocarcinoma, esophageal squamous cell carcinoma, cervical cancer, endometrial cancer, clear cell renal cell carcinoma (ccRCC), and acute myeloid leukemia. Elevated LPCAT1 expression is consistently associated with a poorer prognosis in patients with these cancers [82,83,84,85,86,87,88,89,90,91,92,93,94]. Knockdown studies have demonstrated the essential role of LPCAT1 in the proliferation, migration, invasion, epithelial–mesenchymal transition (EMT) of cancer cells, and overall tumor growth [78,85,90,95,96]. In lung adenocarcinoma, high LPCAT1 expression is linked to the formation of brain metastasis, implicating its involvement in this metastatic process [84]. At least in cervical cancer, the aforementioned effects of LPCAT1 are due to activation of the Janus kinase 2 (JAK2)/signal transducers and activators of the transcription 3 (STAT3) pathway [90]. In contrast, in prostate cancer [78] and oral squamous cell carcinoma [97], LPCAT1 increases the production of the platelet-activating factor (PAF). This factor is responsible for the migration, but not the proliferation, of tumor cells. At the same time, PAF is also responsible for radioresistance; therefore, blocking the activity of LPCAT1 and other enzymes with LPCAT activity increases tumor sensitivity to radiotherapy, particularly in melanoma [98]. In hepatocellular carcinoma, on the other hand, the effect on the expression of S100A11 and Snail may be responsible for the activity of the described enzyme [91,99]. This process depends on the activation of the Wingless-related integration site (Wnt)/β-catenin signaling pathway.

In cutaneous squamous cell carcinoma, LPCAT1 influences the described tumorigenic processes by activating the epidermal growth factor receptor (EGFR) [100]. In esophageal squamous cell carcinoma, the effect of LPCAT1 on tumor processes may be related to an increase in cholesterol synthesis, which is a result of EGFR activation [88]. In endometrial cancer, LPCAT1 affects the activation of the transforming growth factor (TGF)-β-Smad2/3 pathway, which may explain the effect of LPCAT1 on tumor processes in this cancer [92]. LPCAT1 is also responsible for chemoresistance, particularly to paclitaxel in prostate cancer [78] and breast cancer [101] and gefitinib in lung adenocarcinoma [102]. Resistance to gefitinib in lung adenocarcinoma is caused by increased activation of the EGFR-phosphatidylinositol 4,5-bisphosphate 3-kinase (PI3K)-protein kinase B (PKB) signaling pathway by LPCAT1 [102]. The cited data show that LPCAT1 is important in the tumorigenic processes of many types of cancer, which makes it a potential therapeutic target for cancer treatment.

### 4.7. LPCAT2

Lysophosphatidylcholine acyltransferase 2 (LPCAT2), formerly known as AYTL1, AGPAT11, and LPLAT9, exhibits AGPAT activity and substrate specificity for lysophosphatidylserine (lysoPS) and lysoPC [51,103]. Therefore, another name for this enzyme is AGPAT11. LPCAT2 preferentially uses oleoyl-CoA as an acyl group donor [51]. It plays a crucial role in inflammatory responses, localizing to membrane lipid raft domains upon Toll-like receptor 4 activation [104]. This enzyme is also important for the accumulation of lipid droplets, particularly in colorectal cancer cells [105]. Lipid droplets attach to calreticulin (CRT). Consequently, with an increased number of lipid droplets in cancer cells, CRT is not found in the cell membrane, and thus, during drug-induced cancer cell death, it does not activate cytotoxic CD8+ cells [105]. Lipid droplets create a lipophilic environment in cancer cells. In effect, they accumulate lipophilic substances, including many anticancer drugs [106]. This reduces the amount of anticancer drugs in cellular organelles where these drugs exhibit anticancer properties.

For this reason, it is involved in chemoresistance mechanisms, especially against 5-fluorouracil and oxaliplatin. As a consequence, a higher tumor expression of LPCAT2 is associated with a worse prognosis for colorectal cancer patients [105]. In this cancer, LPCAT2 may also play an anticancer role. This enzyme causes ferroptosis [107]; in particular, it causes acetylation of protein arginine methyltransferase 1 (PRMT1), which leads to increased expression of solute carrier family 7 member 11 (SLC7A11), which regulates ferroptosis. Finally, LPCAT2 has potential pro-tumor functions in prostate cancer [108].

### 4.8. LPCAT3

Lysophosphatidylcholine acyltransferase 3 (LPCAT3), previously named MBOAT5, OACT5, and LPLAT12, displays LPCAT activity, with a preference for lysoPC with stearic acid [108,109,110,111,112,113,114,115]. This enzyme can also esterify lysoPS and lysophosphatidylethanolamine (lysoPE), but this activity is much lower than for the esterification of lysoPC. LPCAT3 does not esterify lysoPI and lysoPG [112,113]. LPCAT3 also lacks AGPAT activity [113]. However, another study suggested that it does have such an activity, although not as much as LPCAT [109]. LPCAT3 prefers lysoPC with stearic acid at the sn-1 position [110]. This enzyme can incorporate PUFA, MUFA, and SFA into lysoPC [112,113]. However, there are studies showing that LPCAT3 specifically incorporates PUFA into lysoPC [109,114]. By re-esterifying PUFA, LPCAT3 facilitates the oxidation of cell membrane lipids by ROS. Thus, it is involved in ferroptosis [116,117].

The highest expression of LPCAT3 is found in the intestine and liver, where the enzyme performs its functions [115,118]. The highest expression of LPCAT3 among LPCATs is also found in muscle [119]. In the intestine, LPCAT3 is responsible for lipid absorption [120]. In the liver, LPCAT3 is responsible for the esterification of lysoPC [118]. A decrease in the activity of this enzyme in the liver leads to an increase in the amount of lysoPC in liver cells, which, in turn, leads to an increase in the secretion of apoB-containing lipoproteins [118]. This enzyme is also essential during adipocyte differentiation [114].

LPCAT3 has been implicated in many diseases, particularly insulin resistance in obesity [119,121]. It is the major enzyme with LPCAT activity in adipose tissue [121]. In obesity, there is a higher activity of this enzyme in adipose tissue, as shown by experiments on mice [121]. Furthermore, the highest expression of LPCAT3 among the LPCATs is found in muscle [119]. However, in obese individuals, the expression of this enzyme is increased in muscle compared to lean individuals. This leads to insulin resistance in skeletal muscle. An increase in LPCAT3 expression causes an increase in PUFA incorporation into lysoPC, which consequently alters the function of lipid rafts and thus prevents insulin receptor activation [121]. Overexpression of LPCAT3 in muscle during obesity also leads to myopathy [122]. The same effect of LPCAT3 also occurs in muscle weakness after long periods of inactivity [123]. This may be related to the incorporation of PUFA into phospholipids of the cell membrane, which increases the susceptibility of such phospholipids to peroxidation [123].

Thus, LPCAT3 may, in some ways, protect the body from some of the effects of obesity. In patients with non-alcoholic steatohepatitis (NASH), there is a decrease in LPCAT3 expression in the liver [124], which leads to an increase in the amount of SFA in the inner mitochondrial membrane phospholipids and, consequently, to mitochondrial dysfunction and NASH [124]. Also, in adipose tissue, LPCAT3 reduces the proinflammatory response of this tissue in obesity [125], which is related to the reduction of ROS generation by decreasing the activity of NADPH oxidases.

LPCAT3 expression is increased in osteoarthritis [126]. This is related to the action of interleukin-1β (IL-1β). Increased LPCAT3 expression increases inflammatory responses in the cartilage and, thus, disease progression. LPCAT3 may be involved in tumorigenesis. The expression of this enzyme is upregulated in 16 out of 33 cancer types but also downregulated in 8 types [127]. Higher LPCAT3 expression in acute myeloid leukemia, low-grade glioma, ovarian cancer, and uveal melanoma is associated with a poorer prognosis in patients with these cancers. However, an inverse relationship has been observed in patients with renal clear cell carcinoma [127].

Studies in colon cancer models also suggest that LPCAT3 may inhibit the formation of this cancer [128]. A decrease in LPCAT3 activity leads to an increase in cholesterol production; as cholesterol increases the proliferation of intestinal stem cells, this can lead to intestinal tumors [128].

The first LPCAT3 inhibitors were described in 2022 [116]. As a result, there has been no time to bring them into widespread use. However, LPCAT3 is important in the development of insulin resistance [119,121,129] and in some cancers [127]. For this reason, inhibitors of this enzyme may be potential drugs for these diseases.

### 4.9. LPCAT4

Lysophosphatidylcholine acyltransferase 4 (LPCAT4), also known as AYTL3, LPLAT10, AGPAT7, lysophosphatidylethanolamine acyltransferase (LPEAT)2, and LPAAT-η, localizes to the endoplasmic reticulum and is highly expressed in the uterus, thymus, pancreas, and testes [130]. LPCAT4 is vital for membrane remodeling, introducing DHA into lysophospholipids [131] and thereby increasing the ratio of DHA/arachidonic acid in glycerophospholipids. LPCAT4 is essential for urothelial barrier function [132]. It is associated with DAG and PKC activation.

Its increased expression in hepatocellular carcinoma is associated with a worse prognosis, suggesting a potential role as a therapeutic target for this cancer [18,133]. LPCAT4 is also implicated in cancer, with its expression decreased in colorectal cancer, suggesting an anticancer nature [74]. LPCAT4 activates the WNT-β-catenin pathway in cancer cells, leading to increased cholesterol synthesis, which suggests that LPCAT4 may be a therapeutic target for the treatment of hepatocellular carcinoma.

### 4.10. LCLAT1

LCLAT1, in addition to its AGPAT activity, exhibits LCLAT activity [3,134], particularly towards lysocardiolipin (lysoCL) and other lysopolyglycerophospholipids [135] but not towards lysophospholipids such as lysoPC, lysoPE, and lysoPS [134]. LCLAT1 uses oleoyl-CoA and linoleoyl-CoA as acyl group donors [134]. Other names for LCLAT1 include LPLAT6, ALCAT1, AGPAT8, and lysocardiolipin acyltransferase (LYCAT) [30].

This enzyme is primarily localized to the mitochondria-associated membrane and within mitochondria [3,134], where it plays a crucial role in CL remodeling. Proper functioning of these organelles relies on LCLAT1 activity. However, overexpression of LCLAT1 has been linked to mitochondrial fragmentation and the instability of mitochondrial DNA [134,136].

In the context of cancer, LCLAT1 expression is upregulated in lung cancer, particularly in lung adenocarcinoma. Elevated tumor expression of LCLAT1 is associated with a poorer prognosis in patients with this form of lung cancer [137]. ALCAT1, a variant of LCLAT1, promotes mitochondrial fusion, proliferation, and migration of lung cancer cells and contributes to tumor growth during in vivo experiments [134,137].

LCLAT1 is also implicated in insulin resistance caused by obesity. Studies on mice fed a high-fat diet have demonstrated that a fatty diet upregulates the expression of LCLAT1, leading to changes in CL composition, notably an increase in DHA content. This alteration contributes to oxidative stress, mitochondrial dysfunction, and ultimately insulin resistance [134,138].

Moreover, LCLAT1 has been associated with pulmonary fibrosis [134,139] and Parkinson’s disease [134,140]. The multifaceted roles of LCLAT1 underscore its significance in various physiological processes and diseases.

### 4.11. LPGAT1

Previous names for lysophosphatidylglycerol acyltransferase 1 (LPGAT1) include FAM34A and LPLAT7 [30]. LPGAT1 is localized to the mitochondrial-associated membrane and the endoplasmic reticulum [141,142]. LPGAT1 exhibits LPLAT activity towards lysoPG [141]. As an acyl group donor, LPGAT1 prefers long-chain saturated fatty acyl-CoAs and oleoyl-CoAs [141]. This enzyme also has monoacylglycerol acyltransferase (MGAT) activity [143]. Recent studies have identified another activity of LPGAT1. It can exhibit LPLAT activity, inserting a fatty acid into the *sn*-1 position of lysoPL but not into the *sn*-2 position [130,131]. The acyl group acceptors can be sn-1 lysoPC [144] and *sn*-1 lysoPE [144,145]. As an acyl group donor in this reaction, LPGAT1 prefers stearoyl-CoA [144,145]. LPGAT1 lacks LPLAT activity towards lysoPC, lysoPE, lysoPS, and lysoPI [141]. LPGAT1 also lacks GPAT, diacylglycerol acyltransferase (DGAT), and AGPAT activity [141,143].

The highest expression of LPGAT1 is found in the liver and placenta, with the lowest in the colon. It is responsible for remodeling PG in the mitochondrial-associated membrane [93]. Following this reaction, PG is transported back into the mitochondria, highlighting the essential role of LPGAT1 in mitochondrial function. Additionally, LPGAT1 is involved in TAG synthesis through its MGAT activity [143], particularly crucial in the liver, where its expression is increased in obesity. LPGAT1 is also present in skeletal muscle. Through *sn*-1 activity, LPLAT increases the amount of stearic acid in PC at the *sn*-1 position, which has been observed in skeletal muscle, especially in slow-twitch muscle [146]. This process may play a role in the use of fatty acids as an energy source by this type of muscle.

LPGAT1 is also implicated in various disease states. It is associated with obesity, as demonstrated by studies on *LPGAT1* gene polymorphisms in Native Americans [147] and *LPGAT1* gene knockout mice fed a high-fat diet [142]. The reported association of LPGAT1 with obesity may be related to the MGAT activity of this enzyme [142]. However, LPGAT1 may also be required to maintain homeostasis in a high-fat diet. Reduced LPGAT1 activity leads to insulin resistance in the liver, causing lipid accumulation and hepatosteatosis. Decreased LPGAT1 activity also causes other negative consequences for the liver. It causes lipid accumulation in this organ, which leads to spontaneous hepatosteatosis [142]. In a high-fat diet, reduced LPGAT1 activity leads to hepatofibrosis. LPGAT1 has also been linked to 3-methylglutaconic aciduria with deafness, encephalopathy, and Leigh-like (MEGDEL) syndrome [93,142]. This syndrome is caused by mutations in the genes responsible for PG remodeling, particularly the serine active site containing 1 (*SERAC1)* gene [148], but can also be caused by mutations in this gene.

In cancer, LPGAT1 expression is increased in lung adenocarcinoma tumors, correlating with a worse prognosis for patients with this cancer. The *LPGAT1* gene is part of the five-gene metabolic signatures associated with poor prognosis in patients with lung adenocarcinoma [149]. Studies on lung adenocarcinoma cancer cells with reduced LPGAT1 expression demonstrate its role in tumor cell proliferation, explaining the correlation with prognosis in lung adenocarcinoma patients [149,150].

### 4.12. MBOAT1

Other names for MBOAT1 include OACT1, LPEAT1, and LPLAT14 [30]. MBOAT1 exhibits LPEAT activity [151]. However, another study suggests that MBOAT1 has lysophosphatidylserine acyltransferase (LPSAT) activity [109]. As an acyl group donor, MBOAT1 prefers oleoyl-CoA [109,151] and palmitoyl-CoA [151]. Thus, MBOAT1 plays a role in the Lands cycle remodeling of cell membrane phospholipids, offering protection against ferroptosis to some extent [152]. Additionally, MBOAT1 is crucial for neuronal cell function, supporting neuronal differentiation and neurite outgrowth [151]. Extensive analysis has suggested an association between MBOAT1 and Alzheimer’s disease [153]. Mutations in the MBOAT1 gene lead to brachydactyly syndrome, a developmental disorder [154]. These mutations also cause nonobstructive azoospermia and male infertility [155].

### 4.13. MBOAT2

MBOAT2 exhibits AGPAT and LPEAT activity [109]. It introduces MUFA and linoleic acid into lipids but not arachidonic acid. Other names for MBOAT2 are OACT2 and LPLAT13 [30]. Through its role in modifying the fatty acid composition in phospholipids, MBOAT2 provides some protection against ferroptosis [152].

MBOAT2 is implicated in cancer, with upregulated expression in pancreatic cancer, invasive breast cancer, cholangiocarcinoma, and prostate adenocarcinoma [18,156]. Conversely, it is downregulated in renal cell carcinoma, acute myeloid leukemia, and cutaneous melanoma [18,156]. In pancreatic cancer, higher MBOAT2 expression levels are associated with a worse prognosis due to increased proliferation in cancer cells, as demonstrated in experiments with pancreatic cancer cells.

### 4.14. MBOAT7

MBOAT7, which other names include LPLAT, lysophosphatidylinositol acyltransferase (LPIAT)1, and LPLAT11, shows LPIAT activity [30,109,157]. It can exhibit activity similar to the other lysoPLs, although its enzymatic performance is inferior [157]. MBOAT7 incorporates the arachidonoyl C20:4n-6 and the eicosapentaenoyl C20:5n-3 [109,157]. The MBOAT7-induced enrichment of phosphatidylinositol (PI) with PUFA increases the susceptibility of cells to ferroptosis [158].

MBOAT7 is localized in the endoplasmic reticulum, lipid droplets, and mitochondria-associated membranes [159]. It cooperates in the Lands cycle with acyl-CoA synthetases—in particular, acyl-CoA synthetase long-chain family member (ACSL)3 [160] and ACSL4 [158], enzymes that activate or convert free fatty acids into fatty acyl-CoA [161]. The interaction of ACSL4 and MBOAT7 involves a scaffolding protein associated with monocyte-to-macrophage differentiation (MMD) that facilitates the flow of products of the reactions they catalyze [158]. Thanks to its activity, MBOAT7 participates in the formation of pools of arachidonic acid for the production of eicosanoids [160].

In obesity, MBOAT7 expression is reduced in the liver [162,163,164,165]. MBOAT7 is important in the development of liver diseases, including non-alcoholic fatty liver disease (NAFLD), as evidenced by population-based studies of the rs641738 C > T MBOAT7 genotype. This genotype also correlates with rs8736 T in the MBOAT7 gene [164].

The rs641738 T/T genotype is found in 15–20% of the population, while C/T is found in 40–52% [166,167]. The rs641738 C > T MBOAT7 genotype is a risk factor for fatty liver disease, severe hepatic fibrosis, NAFLD, and hepatocellular carcinoma [168,169,170]. At the same time, there is a synergy between PNPLA3, TM6SF2, and GCKR polymorphisms and MBOAT7 in the likelihood of cirrhosis, severe liver fibrosis, and hepatocellular carcinoma [168,170].

The rs641738 C > T MBOAT7 genotype is associated with decreased MBOAT7 expression in the liver [159,171,172,173]. The reduction in expression of this gene may be due to the close correlation of this genotype with rs8736, which causes an increase in methylation of the MBOAT7 gene and allows the regulation of MBOAT7 mRNA stability by miRNA-24 [164]. Both mechanisms reduce MBOAT7 expression in individuals with the rs8736 T MBOAT7 genotype.

Reduced MBOAT7 expression and activity leads to TAG accumulation in hepatocytes. Several mechanisms for this process have been demonstrated. MBOAT7 is responsible for PI remodeling. Decreased MBOAT7 activity increases the synthesis and degradation of PI to DAG. DAG is then converted to TAG [172]. As MBOAT7 carries out the reaction of PI production from lysoPI, a decrease in the activity of this enzyme causes an increase in the amount of lysoPI in the liver [162]. The compound causes inflammation and liver fibrosis.

The rs8736 T MBOAT7 genotype contributes to a greater inflammatory response, as shown by experiments on macrophages. This is related to an increase in the availability of arachidonic acid for the production of eicosanoids and an increase in the intensity of endoplasmic reticulum stress, which increases Toll-like receptor 4 (TLR4) activity [164].

MBOAT7 also controls the activation of the nucleotide-binding domain leucine-rich repeat containing (NLR) family and pyrin domain containing 3 (NLRP3) inflammasome. Decreased MBOAT7 expression induces an increase in NLRP3 inflammasome activation [164].

Reduced MBOAT7 expression can activate sterol regulatory element-binding protein-1c (SREBP-1c), a transcription factor that upregulates the genes involved in lipid synthesis [174].

Another reason for the increase in TAG accumulation in the liver by decreasing MBOAT7 expression is the increase in the expression of solute carrier family 27 member 1 (SLC27A1)/fatty acid transport protein 1 (FATP1), a protein responsible for the uptake, activation, and channeling of free fatty acids for TAG synthesis [163,175,176]. TAG then accumulates in the form of lipid droplets in liver cells, which facilitates the development of liver disease [170].

A reduction in MBOAT7 expression and activity impairs mitochondrial function in hepatocytes [164,170] and increases their proliferation and migration [170].

All of the aforementioned mechanisms contribute to the increased risk of hepatocellular carcinoma and NAFLD and worsen the course of NAFLD in individuals with the rs641738 C > T MBOAT7 genotype [159,170,171,173]. This genotype is also associated with an increased risk of severe hepatic steatosis in lean individuals without diabetes [177]. Nevertheless, the rs641738 C > T MBOAT7 genotype reduces the likelihood of metabolic syndrome and type 2 diabetes in patients with NAFLD [169,178], although there have been papers that did not find an association of the rs641738 C > T MBOAT7 genotype with the risk of disease and the course of NAFLD [166,179,180].

Reduced MBOAT7 expression in adipose tissue may cause insulin resistance in obesity, as shown by experiments in mice with adipose tissue knockout of this gene [181]. This may explain the association of the rs626283 GG MBOAT7 genotype with insulin resistance in obese Caucasian children [182].

Other studies have shown that the rs641738 C > T MBOAT7 genotype is associated with a worse course of chronic hepatitis B virus (HBV) [183] and hepatitis C virus (HCV) infection [184]. Patients with this genotype have an enhanced inflammatory response and an increased risk of liver fibrosis when chronically infected with these viruses. However, other studies have not confirmed the association of this genotype with the course of HBV or HCV infection [179,185].

MBOAT7 may have some association with alcohol-related liver disease (ALD). Polymorphisms in rs626283 and rs641738 of the MBOAT7 gene are associated with ALD risk [186]. However, the rs641738 C > T MBOAT7 genotype is not associated with liver fibrosis in ALD patients, although it is associated with increased liver inflammation in patients with this liver disease [187]. The reason for the involvement of the reduction in MBOAT7 expression in ALD may be a disturbance in lyso-mal biogenesis in hepatocytes [188].

Significantly, this genotype is not negatively associated with all liver diseases. The rs641738 TT MBOAT7 genotype has been shown to increase survival in male patients with primary sclerosing cholangitis [189].

MBOAT7 is involved in various cancers. It is important in non-small cell lung cancer [160] and clear cell renal cell carcinoma [190]. In the tumors of these cancers, MBOAT7 is upregulated compared to healthy tissues, which is associated with a worse prognosis.

Through the incorporation of arachidonic acid into the PI, MBOAT7 is involved in the production of eicosanoids (Figure 3) [160]. These are lipid mediators involved in tumorigenesis. MBOAT7 also increases the proliferation and migration of cancer cells, as shown in experiments with clear cell renal cell carcinoma cells [190]. However, in some cancers, this downregulation of MBOAT7 expression may be associated with increased tumorigenesis. Hepatocytes with reduced MBOAT7 expression have increased proliferation and migration [170], which may explain the association of reduced MBOAT7 expression, such as with the rs641738 TT MBOAT7 genotype, with an increased risk of hepatocellular carcinoma in NAFLD patients without cirrhosis [171]. On the other hand, data from the GEPIA portal [17] show that higher MBOAT7 expression in hepatocellular carcinoma tumors is associated with worse prognosis [18]. The GEPIA data indicate that MBOAT7 may be an interesting therapeutic target for cancer treatment.

MBOAT7 also plays an important role in brain development (Table 1) [191]. Therefore, mutations in the MBOAT7 gene cause congenital mental retardation with epilepsy [192,193,194,195,196].

The first MBOAT7 inhibitors have recently been developed. These are Sevenin-1 and Sevenin−2 [198]. In the coming years, they will certainly be tested as potential drugs for various liver diseases and cancer.

## 5. Bioinformatics Analysis of the Significance of Enzymes in Cancer

A bioinformatics analysis of the importance of described proteins in cancer processes was performed on the GEPIA portal [17,18] that analyzes raw gene expression data from nearly 10,000 tumor samples from 33 cancer types obtained from The Cancer Genome Atlas (TCGA) [199].

The GEPIA allows the analysis of:-the expression level of the selected gene in the tumor relative to adjacent tumor tissue and healthy tissue,-analysis of the association between the expression level of the selected gene in the tumor and the prognosis of patients with the selected cancer type,-the correlation between the expression of two genes in the tumor of selected cancer types.

Correlation of the prognosis of patients with the level of expression of genes in the tumor best reflects the importance of these genes in cancer processes. Therefore, such an analysis was performed for the genes discussed in this paper, and the results are summarized in Table 2 and Table 3. Differences in the overall survival between cases with the highest and lowest quartiles of tumor expression of the selected gene were analyzed. The results were considered statistically significant when the Mantel–Cox test yielded a *p*-value of 0.05 or less. However, *p*-values between 0.05 and 0.10 were considered worth reporting as a trend towards a better or worse prognosis. For this reason, such results are also included in the published tables.

Bioinformatics analysis showed that only some genes discussed in this paper are associated only with a worse prognosis. There are three such genes: *AGPAT4*, *LPCAT4*, and *MBOAT7*. *AGPAT4* is associated with a worse prognosis in nine cancer types and with a trend towards a worse prognosis in two cancer types (0.05 < *p* < 0.10), *LPCAT4* in five cancer types and in two cancer types with a trend towards a worse prognosis, and *MBOAT7* in seven cancer types and in four cancer types with a trend towards a worse prognosis. This suggests that drugs targeting the proteins encoded by these three genes may be, in a sense, universal anticancer drugs or at least act on many types of cancer.

Other genes that are associated with a worse prognosis in most cancers, but also with better prognosis in some cancers, are *AGPAT2*, *MBOAT2*, *LPGAT1*, *AGPAT1*, and *LPCAT2*. *AGPAT2* is associated with a worse prognosis in six cancer types and with a better prognosis in one type; *AGPAT1*, *LPGAT1*, and *MBOAT2* with a worse prognosis in five types and one with a better prognosis; and LPCAT2 with a worse prognosis in five types and two with a better prognosis. The proteins encoded by these six genes may be proteins with pro-cancer properties that act on many types of cancer. At the same time, these proteins are associated with a better prognosis in some cancers. Therefore, in order to understand the effects of drugs targeting these proteins, studies are needed to show which anticancer mechanisms these three proteins are involved in.

In certain types of cancer, many of the genes studied are associated with a poorer prognosis. This suggests that lipid metabolism is important for tumorigenesis in these cancers. These cancers are hepatocellular carcinoma (10 genes associated with a worse prognosis out of 21 analyzed), lower-grade glioma (11 genes with a worse prognosis and 1 with a better prognosis), and adrenocortical carcinoma (6 genes with a worse prognosis). This suggests that, in these three cancer types, many sites of phospholipid and TAG biosynthesis are relevant to tumorigenic processes. Therefore, targeting lipid metabolism during therapy may be therapeutic in these three cancers.

Bioinformatic analysis also showed that, in ccRCC, 12 genes out of 21 analyzed were associated with a better prognosis and 2 with a worse prognosis. This suggests that, in this type of cancer, phospholipid and TAG biosynthesis are involved in anti-tumor processes at multiple stages. However, more detailed studies of the relationships found in this type of cancer are needed.

Bioinformatics analysis has also shown that certain analyzed genes are often associated with a better prognosis. An example of such a gene is *GPAT4* (in two cancer types associated with a worse prognosis, in four with a better one, and a trend towards a worse prognosis in two). This shows that *GPAT4* may have anticancer properties, at least in several types of cancer.

## 6. Conclusions

Our article thoroughly explains the catalytic functions and roles of lipid metabolism proteins in cell processes. However, their involvement in diseases, especially obesity and cancer, is not as well explored. A basic bioinformatics analysis suggests potential as therapeutic targets for some of the discussed proteins. Yet, as specific inhibitors are largely unknown for the vast majority of enzymes, it seems crucial to develop and test these inhibitors in the coming years, focusing on their potential anticancer properties under in vitro and in vivo conditions.

## Figures and Tables

**Figure 1 cancers-16-02115-f001:**
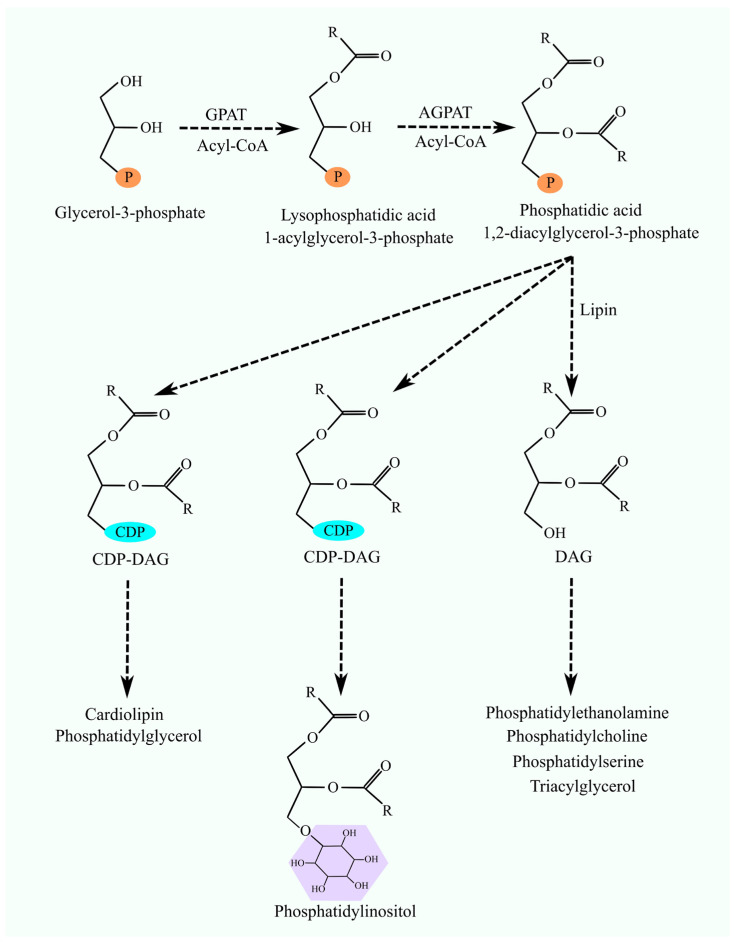
Synthesis of glycerophospholipids and TAG. The initial stages of the synthesis of all described glycerophospholipids and TAG involve attaching two acyl groups to glycerol-3-phosphate. The addition of the first acyl group is catalyzed by GPAT and the second one by AGPAT, leading to the formation of phosphatidic acid. Subsequently, the lipid synthesis pathway diverges into three routes. In the first route, phosphatidic acid is transformed into DAG by lipins. DAG then undergoes further reactions to become PE, PC, PS, or TAG. Phosphatidic acid can also be converted into CDP-DAG. CDP-DAG is transformed into PI or CL and PG.

**Figure 2 cancers-16-02115-f002:**
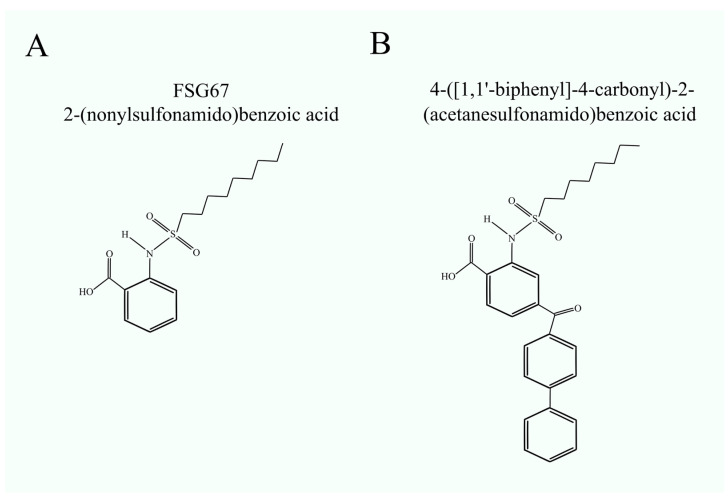
GPAT inhibitors. Structural formulas of (**A**) FSG67 (2-(nonylsulfonamido)benzoic acid) and (**B**) 4-([1,1′-biphenyl]-4-carbonyl)-2-(acetanesulfonamido)benzoic acid. FSG67 is the most studied GPAT inhibitor. In contrast, the second compound shown was developed based on in silico studies of previously developed inhibitors, including FSG67.

**Figure 3 cancers-16-02115-f003:**
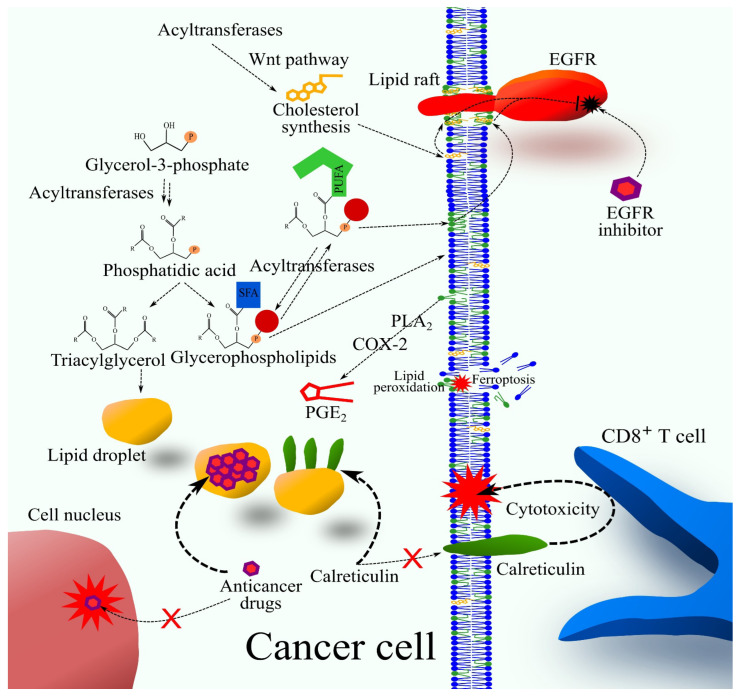
The significance of phospholipid acyltransferases for cancer cells. Phospholipid acyltransferases are enzymes in the synthesis pathway of phospholipids and TAG. The increased activity of certain phospholipid acyltransferases (GPAM, GPAT3, GPAT4, AGPAT2, and MBOAT7) boosts TAG production. After synthesis, TAG form lipid droplets. These enzymes (LPCAT1, LPCAT2, and MBOAT7) can also indirectly influence lipid droplet biogenesis. Lipid droplets are involved in chemoresistance, as they can accumulate lipophilic substances, including many anticancer drugs; consequently, these drugs are not present in the organelles where they exhibit anticancer effects. Lipid droplets also bind calreticulin, preventing this protein from being found in the cell membrane. When cancer cells die due to anticancer drugs, calreticulin from the cell membrane activates cytotoxic CD8+ T cells. Reduced calreticulin in the cell membrane disrupts this mechanism. Glycerophospholipids are also involved in cancer processes as structural elements of cells. Therefore, cancer cells, which divide frequently, must exhibit increased glycerophospholipid synthesis, hence the activity of phospholipid acyltransferases. Some phospholipid acyltransferases incorporate PUFAs into glycerophospholipids (AGPAT3, AGPAT4, AGPAT5, LPCAT3, and LPCAT4). However, not all acyltransferases in the Lands cycle incorporate PUFAs into phospholipids. For example, LPCAT1, MBOAT1, and MBOAT2 incorporate oleic acid into the sn-2 position of phospholipids. Phospholipids can contain arachidonic acid, which can be released by PLA2 and processed into PGE2 by cyclooxygenase-2 (COX-2). This bioactive lipid is involved in cancer processes. Phospholipids containing PUFAs can also undergo peroxidation. Therefore, increased activity of phospholipid acyltransferases, which modify the fatty acid composition in phospholipids, changes the susceptibility of cancer cells to ferroptosis. Phospholipid acyltransferases affect cancer cell metabolism, increasing cholesterol production through the Wnt pathway. Cholesterol and PUFA-containing phospholipids form lipid rafts, influencing membrane receptors that require lipid rafts for their activity, such as EGFR. Thus, phospholipid acyltransferases, by directly producing PUFA-containing phospholipids and indirectly increasing cholesterol synthesis, enhance EGFR activation and weaken the effect of EGFR inhibitors. This is a mechanism of chemoresistance against EGFR inhibitors.

**Table 1 cancers-16-02115-t001:** Characteristics of AGPAT and LPLAT.

References	Involvement in Diseases	Physiological Significance	Donor Preference of the Acyl Group	Activity	Official Name of the Gene (Other Names *)
[41,53]		Myoblast differentiation	palmitoyl-CoA, oleoyl-CoA, linoleoyl-CoA,	AGPAT, much smaller LPLAT	*AGPAT1* (LPAAT-α, LPLAT1)
[53,54,55,56,57,58,59,60,61,62,63,64]	Cancers—breast cancer, ovarian cancer, osteosarcoma, prostate cancer (pro-tumor properties); mutations in the *AGPAT2* gene cause BSCL1	Adipogenesis	oleoyl-CoA, linoleoyl-CoA	AGPAT, much smaller LPLAT	*AGPAT2* (LPAAT-β, LPLAT2)
[65,66,67,68,69,70,71]	Cancers—colorectal cancer, gastric cancer (anti-tumor properties); autism spectrum disorder. Mutations in the *AGPAT3* gene are associated with IDRP syndrome	Spermatogenesis, skeletal muscle physiology, neuronal migration, adipogenesis	oleoyl-CoA, DHA-CoA	AGPAT, LPLAT	*AGPAT3* (LPAAT3, LPAAT-γ, LPLAT3)
[70,72,73]	Cancers—colorectal cancer (pro-tumor properties)	Incorporation of DHA into phospholipids in the brain	DHA-CoA	AGPAT	*AGPAT4* (LPAAT4, LPAAT-δ, LPLAT4)
[49,74]	Colorectal cancer (anti-tumor properties)		oleoyl-CoA (AGPAT), DHA-CoA (LPLAT)	AGPAT, LPLAT	*AGPAT5* (LPAAT-ε, LPLAT5)
[50,75,76,77,78,79,80,81,82,83,84,85,86,87,88,89,90,91,92,93,94,95,96,97,98,99,100,101,102]	Cancers—breast cancer, cervical cancer, esophageal squamous cell carcinoma, endometrial cancer, head and neck squamous cell carcinoma, hepatocellular carcinoma, kidney cancer (clear cell renal cell carcinoma), leukemia (acute myeloid leukemia), lung cancer (lung squamous cell carcinoma, lung adenocarcinoma), skin cancer (cutaneous squamous cell carcinoma; melanoma) (pro-tumor properties); psoriasis	Activity decreases sensitivity to ferroptosis. Defense by PUFA influence. Increasing transcription by causing histone H4 palmitoylation. Involvement in the physiology of lipid droplets	oleoyl-CoA	AGPAT, LPCAT, LPGAT	*LPCAT1* (AGPAT9, AGPAT10, LPLAT8)
[51,103,104,105,106,107,108]	Cancers—colorectal cancer, prostate cancer (pro-tumor properties)	Effects on TLR4	oleoyl-CoA	AGPAT, LPCAT	*LPCAT2* (AYTL1, AGPAT11, LPLAT9)
[109,110,111,112,113,114,115,116,117,118,119,120,121,122,123,124,125,126,127,128,129]	Insulin resistance in obesity; obesity-related myopathy; NASH; osteoarthritis; Cancers—brain tumor (low-grade glioma), colon cancer, leukemia (acute myeloid leukemia), ovarian cancer, melanoma (uveal melanoma) (pro-tumor properties) renal clear cell carcinoma (anti-tumor properties); muscle weakness caused by inactivity	Activity increases sensitivity to ferroptosis. Lipid absorption. LysoPC esterification in the liver. Adipocyte differentiation.	PUFA-CoA	LPCAT, LPEAT, LPSAT	*LPCAT3* (MBOAT5, OACT5, LPLAT12)
[18,74,130,131,132,133]	Cancers—hepatocellular carcinoma (pro-tumor properties)	Urothelial barrier function	DHA-CoA	LPLAT	*LPCAT4* (AYTL3, LPLAT10, AGPAT7, LPEAT2, LPAAT-η)
[134,135,136,137]	Cancers—lung adenocarcinoma (pro-tumor properties); insulin resistance in obesity; pulmonary fibrosis; Parkinson’s disease	CL remodeling in mitochondria	oleoyl-CoA, linoleoyl-CoA	AGPAT, LCLAT	*LCLAT1* (LPLAT6, ALCAT1, AGPAT8, LYCAT)
[93,141,142,143,144,145,146,147,148,149,150]	Cancer—lung adenocarcinoma (pro-tumor properties); obesity; MEGDEL syndrome	PG remodeling, TAG synthesis in liver, skeletal muscle physiology	palmitoyl-CoA, stearoyl-CoA oleoyl-CoA	LPGAT, MGAT, sn-1 LPCAT, sn-1 LPEAT.	*LPGAT1* (FAM34A, LPLAT7)
[109,151,152,153,154,155]	Alzheimer’s disease; mutations in the gene cause brachydactyly-syndactyly syndrome and nonobstructive azoospermia	Protects against ferroptosis. Physiology of nerve cells	palmitoyl-CoA, oleoyl-CoA	LPEAT, LPSAT	*MBOAT1* (OACT1, LPEAT1, LPLAT14)
[18,109,152,156]	Cancers—pancreatic cancer (pro-tumor properties) invasive breast cancer, cholangiocarcinoma, prostate adenocarcinoma (pro-tumor properties?) leukemia (acute myeloid leukemia), kidney cancer (renal cell carcinoma), skin cancer (cutaneous melanoma) (anti-tumor properties?)	Protects against ferroptosis	MUFA-CoA linoleoyl-CoA	AGPAT, LPEAT	*MBOAT2* (OACT2, LPLAT13)
[18,109,157,158,159,160,161,162,163,164,165,166,167,168,169,170,171,172,173,174,175,176,177,178,179,180,181,182,183,184,185,186,187,188,189,190,191,192,193,194,195,196]	Reduced expression and activity are important for increased risk and worse outcomes in liver disease: NAFLD, ALD, chronic HBV or HCV infection. Lower expression associated with increased risk of hepatocellular carcinoma. Associated with insulin resistance in obesity. Involved in certain cancers—hepatocellular carcinoma, kidney cancer (clear cell renal cell carcinoma), lung cancer (non-small cell lung cancer) (pro-tumor properties) Mutations in the *MBOAT7* gene cause congenital mental retardation with epilepsy.	PI remodeling, introduction of arachidonic acid into this phospholipid. Brain development	arachidonoyl-CoA, EPA-CoA	LPIAT	*MBOAT7* (LPLAT, LPIAT1, LPLAT11)

* According to the HUGO Gene Nomenclature Committee [30,197].

**Table 2 cancers-16-02115-t002:** Association of GPAT and AGPAT activity enzyme expression with the overall survival for patients with selected cancers. ↓, red background—higher expression is associated with a worse prognosis for a patient with a given cancer; ↑, blue background—higher expression is associated with a better prognosis for a patient with a given cancer; -, gray background—expression of a particular gene is not significantly associated with the patient prognosis.

Name of the Cancer	*GPAM*	*GPAT2*	*GPAT3 (AGPAT9)*	*GPAT4 (AGPAT6)*	*AGPAT1*	*AGPAT2*	*AGPAT3*	*AGPAT4*	*AGPAT5*
Adrenocortical carcinoma	-	-	↑	-	↓	↓	-	↓	-
Bladder urothelial carcinoma	-	-	↓	-	-	-	-	↓	-
Breast invasive carcinoma	-	↑ *p* = 0.079	-	↓	↓	↓ *p* = 0.072	-	-	↓ *p* = 0.10
Cervical squamous cell carcinoma and endocervical adenocarcinoma	-	↑ *p* = 0.070	-	-	-	-	↓	↓	-
Cholangiocarcinoma	-	-	-	-	-	-	-	-	-
Colon adenocarcinoma	-	-	-	↑	-	-	-	-	↑ *p* = 0.058
Lymphoid neoplasm diffuse large B-cell lymphoma	-	-	-	-	-	↑ *p* = 0.079	-	-	-
Esophageal carcinoma	-	-	-	-	-	-	-	-	-
Glioblastoma multiforme	-	-	↓	-	-	↓ *p* = 0.075	-	-	-
Head and neck squamous cell carcinoma	-	-	-	↑	-	↓	-	↓	-
Kidney chromophobe	-	-	-	-	-	↑	-	-	-
Kidney renal clear cell carcinoma	↑	-	↑	↑	↑	-	↑	-	↑
Kidney renal papillary cell carcinoma	-	-	-	-	-	-	-	-	-
Acute myeloid leukemia	-	↓	↓	-	↓ *p* = 0.093	↓ *p* = 0.097	↓	-	-
Brain lower grade glioma	-	↓	↓	-	↑ *p* = 0.095	↓	-	-	-
Liver hepatocellular carcinoma	-	-	-	-	↓	↓ *p* = 0.10	-	↓	↓
Lung adenocarcinoma	-	-	↓ *p* = 0.051	-	-	-	-	-	-
Lung squamous cell carcinoma	-	-	-	-	-	↓	-	-	-
Mesothelioma	-	-	↑	↓	↑ *p* = 0.083	-	-	↓	↓
Ovarian serous cystadenocarcinoma	-	-	-	↓ *p* = 0.079	-	-	-	↓ *p* = 0.080	-
Pancreatic adenocarcinoma	-	-	-	-	-	-	-	-	-
Pheochromocytoma and Paraganglioma	-	-	↓	-	-	-	-	-	-
Prostate adenocarcinoma	-	-	-	-	-	-	-	-	-
Rectum adenocarcinoma	-	-	-	-	-	-	↑	-	-
Sarcoma	-	-	-	-	-	-	-	-	↓ *p* = 0.069
Skin cutaneous melanoma	-	-	-	↓ *p* = 0.062	↓	↓	↓	↓ *p* = 0.076	-
Stomach adenocarcinoma	-	-	-	-	↓ *p* = 0.089	↑ *p* = 0.064	-	↓	-
Testicular germ cell tumors	-	-	-	-	-	-	-	-	-
Thyroid carcinoma	-	-	-	-	↓	-	-	↓	-
Thymoma	-	-	-	-	-	-	↓ *p* = 0.085	-	-
Uterine corpus endometrial carcinoma	↑ *p* = 0.082	-	-	-	-	-	-	-	-
Uterine carcinosarcoma	-	-	-	-	-	↓	-	-	-
Uveal Melanoma	-	-	-	↑	-	↑ *p* = 0.061	↓	↓	-

**Table 3 cancers-16-02115-t003:** Association of the expression of enzymes with LPLAT activity with the overall survival for patients with selected cancers. ↓, red background—higher expression is associated with a worse prognosis for a patient with a given cancer; ↑, blue background—higher expression is associated with a better prognosis for a patient with a given cancer; gray background—expression of a particular gene is not significantly associated with the patient prognosis.

Name of the Cancer	*LPCAT1*	*LPCAT2*	*LPCAT3*	*LPCAT4*	*LCLAT1*	*LPGAT1*	*MBOAT1*	*MBOAT2*	*MBOAT7*
Adrenocortical carcinoma	-	-	-	↓	-	↓	-	↓	-
Bladder urothelial carcinoma	-	↓ *p* = 0.10	-	-	-	-	-	↓	-
Breast invasive carcinoma	↓ *p* = 0.051	-	-	-	↓ *p* = 0.052	↓ *p* = 0.068	-	-	↓
Cervical squamous cell carcinoma and endocervical adenocarcinoma	↓	↓ *p* = 0.051	-	-	-	-	-	↓ *p* = 0.077	↓ *p* = 0.076
Cholangiocarcinoma	-	-	-	-	-	-	-	-	-
Colon adenocarcinoma	-	-	-	-	-	-	-	-	-
Lymphoid neoplasm diffuse large B-cell lymphoma	-	-	-	-	-	↓ *p* = 0.087	-	-	-
Esophageal carcinoma	-	-	-	-	-	-	-	-	-
Glioblastoma multiforme	-	-	-	-	-	-	-	-	-
Head and neck squamous cell carcinoma	-	↓	-	-	↓	-	-	-	-
Kidney chromophobe	-	-	-	-	-	↓	-	-	-
Kidney renal clear cell carcinoma	-	-	↑	↓	↑	↑	↑	↑	↓ *p* = 0.063
Kidney renal papillary cell carcinoma	↓ *p* = 0.083	-	↑ *p* = 0.074	↓ *p* = 0.060	-	↓	-	-	↓
Acute myeloid leukemia	↓ *p* = 0.055	↑ *p* = 0.089	↓	-	-	-	-	-	-
Brain lower grade glioma	↓	↓	↓	-	↓ *p* = 0.063	↓	↓	-	↓
Liver hepatocellular carcinoma	↓	↓	-	↓	↓	-	↓	-	↓
Lung adenocarcinoma	↑	-	-	-	↓ *p* = 0.054	↓	-	-	↓
Lung squamous cell carcinoma	↓ *p* = 0.061	-	-	-	-	-	-	-	↓ *p* = 0.051
Mesothelioma	-	-	-	-	↓	-	-	↓	↓
Ovarian serous cystadenocarcinoma	-	-	↓	-	-	-	-	-	↓ *p* = 0.053
Pancreatic adenocarcinoma	↑	↓	-	↓	-	-	↓	↓	-
Pheochromocytoma and Paraganglioma	-	-	-	-	-	-	-	-	-
Prostate adenocarcinoma	↓ *p* = 0.076	-	-	-	-	-	-	-	-
Rectum adenocarcinoma	-	-	-	-	-	-	↑	-	-
Sarcoma	-	-	-	-	↓ *p* = 0.075	-	-	-	-
Skin cutaneous melanoma	-	↑	-	-	-	-	-	-	-
Stomach adenocarcinoma	-	-	-	-	-	-	↑ *p* = 0.067	-	-
Testicular germ cell tumors	-	-	-	-	-	-	-	-	-
Thyroid carcinoma	-	-	-	-	↓ *p* = 0.097	-	-	-	↓
Thymoma	↑ *p* = 0.10	↓	↑ *p* = 0.084	-	-	↓ *p* = 0.062	-	-	-
Uterine corpus endometrial carcinoma	-	↑	-	-	-	-	-	-	-
Uterine carcinosarcoma	-	-	-	-	-	-	-	-	-
Uveal Melanoma	↓	-	-	↓	-	-	-	↓	-

## Data Availability

Not applicable.

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
