# Peer review of "Phospholipid Acyltransferases: Characterization and Involvement of the Enzymes in Metabolic and Cancer Diseases"

_cancers, 2024, doi:10.3390/cancers16112115_

Round 1

Reviewer 1 Report

Comments and Suggestions for Authors

Comments and Suggestions to Authors

In this article, Korbecki et al. discuss the enzymatic processes governing the initial stages of synthesizing glycerophospholipids and triacylglycerol. They further discuss GPAT, AGPAT and LPLAT as key enzymes in the review. The authors further describe the potential therapeutic implications of inhibiting the enzymes in treating metabolic diseases and cancer. The study could be a good addition to the literature but needs further revision.

1.     The manuscript's language needs to be improved by a native speaker.

2.     Please define abbreviations when they first appear in the manuscript.

3.     The abstract needs revisions as per the revisions.

4.     Please update the manuscript with recent literature in 2023-2024.

5.     Please revise the graphical abstract using biorender for better quality and presentation.

6.     L110, please revise the statement and remove that link; use it in the reference, but keep the citation with text.

7.     L195, Figure 2. Please also add the names of the inhibitors with the structures in the illustration.

8.     L240, Table 1. Please add another column for references and keep this consistent in all tables.

9.     L241, move the link to the reference in the bibliography and keep the citation here; please ensure consistency throughout the manuscript.

  1. L514, In pancreatic cancer….., missing citation.
  2. L517, A study from NC showed that increased MBOAT7 expression is linked to hepatocellular and renal cancers 10.1038/s41467-023-38932-5
  3. Tables 2 and 3's down (red background) and up (blue background) arrows are confusing to readers, as per the description provided on the table titles. The down arrow depicts down-regulation and, likewise, decreased expression. The same concept goes with the up arrow. How can authors address it better to support the understanding of the journal’s readers?
  4. Please draw a detailed illustration using the biorender of phospholipid acyltransferases crosstalk in signaling pathways involved in cancer metabolism and therapy.

14.  Please format references as per the journal’s instructions.

Comments on the Quality of English Language

1. The manuscript's language needs to be improved by a native speaker.

Author Response

  1. The manuscript's language needs to be improved by a native speaker.

In accordance with the Reviewer's comment, the manuscript has been corrected by a native speaker, we attach a certificate.

  1. Please define abbreviations when they first appear in the manuscript.

According to Reviewer’s remark the entire article has been revised to include abbreviations.

  1. The abstract needs revisions as per the revisions.

According to Reviewer’s remark abbreviations in the abstract have been corrected.

  1. Please update the manuscript with recent literature in 2023-2024.

According to Reviewer’s remark the article has been updated with the latest articles

  1. Please revise the graphical abstract using biorender for better quality and presentation.

According to Reviewer’s remark graphical abstract has been corrected, but we did not use the Biorender website due to the phospholipid chemical structure we drew.

  1. L110, please revise the statement and remove that link; use it in the reference, but keep the citation with text.

Corrected according to the Reviewer's recommendation. Website citations introduced

  1. L195, Figure 2. Please also add the names of the inhibitors with the structures in the illustration.

According to Reviewer’s remark inhibitor names have been added.

  1. L240, Table 1. Please add another column for references and keep this consistent in all tables.

According to Reviewer’s remark references have been added.

  1. L241, move the link to the reference in the bibliography and keep the citation here; please ensure consistency throughout the manuscript.

Corrected according to the Reviewer's recommendation. Website citations introduced.

  1. L514, In pancreatic cancer….., missing citation.

According to Reviewer’s remark references have been added.

  1. L517, A study from NC showed that increased MBOAT7 expression is linked to hepatocellular and renal cancers 10.1038/s41467-023-38932-5. 

The mentioned article cites 2 articles about the importance of MBOAT7 in hepatocellular and renal cancers. We cite these two articles in our review.

  1. Tables 2 and 3's down (red background) and up (blue background) arrows are confusing to readers, as per the description provided on the table titles. The down arrow depicts down-regulation and, likewise, decreased expression. The same concept goes with the up arrow. How can authors address it better to support the understanding of the journal’s readers?

The table description has been corrected.

  1. Please draw a detailed illustration using the biorender of phospholipid acyltransferases crosstalk in signaling pathways involved in cancer metabolism and therapy.

According to Reviewer’s remark the figures has been made.

  1. Please format references as per the journal’s instructions.

According to Reviewer’s remark the bibliography has been corrected.

Reviewer 2 Report

Comments and Suggestions for Authors

Dear Authors

Thank you for the opportunity to review your work.

I found the theme very interesting and in line with oncobiology studies focused not only in mutations but in metabolic alterations involved in carcinogenesis and tumor progression which can  lead to new therapeutic targets for this group of diesases which represents a very high disease burden globally. Namely, it is being increasing accepted that alterations in lipid metabolism is important in carcinogenesis.

The work is really interesting and a very comprehensive review on the role of phosplipid acyltransferases in metabolic diseases such as obesity and diabetes and in inumerous cancer types. However, because of this axaustive review it is also sometimes difficult to follow: 

1. I don´t think the graphical abstract is in line with the text. It is importnat that you review and improve it according to your work.

2. Because there are so many enzymes being addressed, and each can have different nomeclatures the reading can be a bit tiring. For instance in section 2 when you address Glycerol-3-phosphate acyltransferases  (GPAT) you start talking about the diferent isoforms and state that GPAM is another name for GPAT1. However, in the subsequent text you keep addressing this enzyme as GPAM. I think it would simplify if you use GPAT1 as the other isoforms all use this nomeclature (GPAT2, GPAT3, GPAT4)... The same occurs for other enzymes during the text...

3. I do not know if Table 1 is really needed: all information within is included, with additional explanations in the text...

4.  Tables 2 and 3 are very interesting giving a visual overview of the GEPIA analysis and conclusions for several different cancer types and can be important for the understanding of the role of enzyme expression and distinct tumor progressin and patient prognosis. This has been done recurring to a very elegant bioinformatics approach that has led to very interesting insights on the relationship between different enzymes gene expression and patient prognosis, hopefully leading to his can open opportunities for viewing these enzymes as therapeutic targets...

5. The phrase in line 650 "...may be anticancer drugs..." should be altered to "...may be proteins with anti-cancer properties..."

Author Response

Rev.1.

  1. I don´t think the graphical abstract is in line with the text. It is importnat that you review and improve it according to your work.

Graphical abstract has been improved according to Reviewer's recommendations

  1. Because there are so many enzymes being addressed, and each can have different nomeclatures the reading can be a bit tiring. For instance in section 2 when you address Glycerol-3-phosphate acyltransferases  (GPAT) you start talking about the diferent isoforms and state that GPAM is another name for GPAT1. However, in the subsequent text you keep addressing this enzyme as GPAM. I think it would simplify if you use GPAT1 as the other isoforms all use this nomeclature (GPAT2, GPAT3, GPAT4)... The same occurs for other enzymes during the text...

All enzyme names and abbreviations used were taken from the HUGO Gene Nomenclature Committee (HGNC) website (https://www.genenames.org/). These are the official names of the genes.

Seal RL, Braschi B, Gray K, Jones TEM, Tweedie S, Haim-Vilmovsky L, Bruford EA. Genenames.org: the HGNC resources in 2023. Nucleic Acids Res. DOI: 10.1093/nar/gkac888

  1. I do not know if Table 1 is really needed: all information within is included, with additional explanations in the text...

Table 1 is a summary of the section of the article discussing all acylglycerol-3-phosphate acyltransferases and lysophospholipids acyltransferases. For this reason, some readers will find this table helpful in understanding the topic of the article. For this reason, we would like to ask the Reviewer to leave table 1 in the article.

  1. Tables 2 and 3 are very interesting giving a visual overview of the GEPIA analysis and conclusions for several different cancer types and can be important for the understanding of the role of enzyme expression and distinct tumor progressin and patient prognosis. This has been done recurring to a very elegant bioinformatics approach that has led to very interesting insights on the relationship between different enzymes gene expression and patient prognosis, hopefully leading to his can open opportunities for viewing these enzymes as therapeutic targets...

  1. The phrase in line 650 "...may be anticancer drugs..." should be altered to "...may be proteins with anti-cancer properties..."

The text has been corrected in accordance with the Reviewer's recommendations.

Reviewer 3 Report

Comments and Suggestions for Authors

The manuscript of Korbecki et al. deals with a review article on the enzymatic processes governing the initial stages of triglycerides and glycerol-phospholipids. The review begins by discussing the properties of these enzymes, emphasizing their specificity in enzymatic reactions, notably the incorporation of poly-unsaturated fatty acids (PUFA) such as arachidonic acid and docosahexaenoic acid (DHA) into phospholipids. Notably, the paper sheds light on the intricate involvement of these enzymes in various diseases, including obesity, insulin resistance, and cancer. As a result of this review, authors provide a comprehensive understanding of these processes and their potential as therapeutic targets. Present data could be useful in the study of metabolic diseases including cancers, and the manuscript seems to be adequately written. From here, this review could be suitable for publication in its present form.

Author Response

We would like to thank Reviewer for for reliable and solid review.

Reviewer 4 Report

Comments and Suggestions for Authors

The review by Korbecki et al. provides a comprehensive function view of the glycerol-3-phosphate acyltransferases (GPAT), the 1-acylglycerol-3-phosphate acyltransferases (AGPAT) and lysophospholipid acyltransferase (LPLAT) family enzymes and their potential role in obesity and cancer. While not well consolidated/documented, this can be understandable due to the lack of studies involving these enzymes in cancer. The review can be considered as a useful opinion for future research direction in the field. Nevertheless, authors have summarized the gene expression in cancers by extracting data from GEPIA portal, and this can be helpful for investigators in the field. Also, they provided some information on the only one inhibitor available, thus opening discussion on the opportunity of developing inhibitors and exploring mechanisms of these enzymes in cancer.

Comments for improvement.

-       Can you describe studies that have characterized the enzymatic inhibitory effect of the inhibitor FSG67, what is the IC50 and whether there is evidence on pharmacodynamics of this inhibitor that make it appropriate for in vivo studies.

-       The table 1 is very interesting and helpful, please add citations by inserting (new column) the appropriate reference describing the role these genes in the described disease.

Author Response

Comments for improvement.

-       Can you describe studies that have characterized the enzymatic inhibitory effect of the inhibitor FSG67, what is the IC50 and whether there is evidence on pharmacodynamics of this inhibitor that make it appropriate for in vivo studies.

In accordance with the Reviewer's recommendations, relevant fragments have been added.

-       The table 1 is very interesting and helpful, please add citations by inserting (new column) the appropriate reference describing the role these genes in the described disease.

In accordance with the Reviewer's recommendations references have been added.